# Use of Steaming Process to Improve Biochemical Activity of *Polygonatum sibiricum* Polysaccharides against D-Galactose-Induced Memory Impairment in Mice

**DOI:** 10.3390/ijms231911220

**Published:** 2022-09-23

**Authors:** Zhijuan Bian, Congting Li, Daiyin Peng, Xuncui Wang, Guoqi Zhu

**Affiliations:** Key Laboratory of Xin’an Medicine, the Ministry of Education, Key Laboratory of Molecular Biology (Brain Diseases), Anhui University of Chinese Medicine, Hefei 230012, China

**Keywords:** aging, memory impairment, steaming, synaptic, oxidative stress

## Abstract

Polysaccharide from *Polygonatum sibiricum* (PSP) possesses antioxidant, antiaging, and neuroprotective activities. However, whether and how the steaming process influences the biological activities of PSP, especially against aging-related memory impairment, is not yet known. In this study, *Polygonatum sibiricum* rhizome was subjected to a “nine steaming and nine drying” process, then PSPs with different steaming times were abstracted. Thereafter, the physicochemical properties were qualified; the antioxidant activities of PSPs were evaluated in a D-gal-induced HT-22 cell model, and the effects of PSPs (PSP0, PSP5 and PSP9) on memory was evaluated using D-gal-injured mice. Our results showed that while the steamed PSPs had a low pH value and a large negative charge, they shared similar main chains and substituents. Cellular experiments showed that the antioxidant activity of steamed PSPs increased. PSP0, PSP5, and PSP9 could significantly ameliorate the memory impairment of D-gal-injured mice, with PSP5 showing the optimal effect. Meanwhile, PSP5 demonstrated the best effect in terms of preventing cell death and synaptic injury in D-gal-injured mice. Additionally, the steamed PSPs increased anti-oxidative stress-related protein expression and decreased inflammation-related protein expression in D-gal-injured mice. Collectively, the steaming process improves the effects of PSPs against D-gal-induced memory impairment in mice, likely by increasing the antioxidant activity of PSPs.

## 1. Introduction

Nowadays, the average life expectancy of human beings has been greatly improved. The global elderly population is expected to reach 2.1 billion by 2050 [1]. Effectively alleviating aging is thus of great significance for the current situation of population aging [2]. In addition, the aging process is associated with a general decline in various aspects of cognition and brain function; study has shown that functional decline in spatial cognition is strongly correlated with aging [3].

The pathological changes involved in aging have a particularly profound effect on cerebral functioning, as the brain is most severely damaged by impairments in oxygen demand, regenerative capacity, and endogenous antioxidant capacity [4]. The hippocampus, an important brain region, plays a crucial role in memory, navigation, and cognition [5,6]. Aging-related cognitive impairment is associated with structural changes in the brain, such as reductions in cortical thickness and hippocampal volume [7]. Oxidative stress and inflammatory dysfunction cause long-term learning and memory decline during the aging process, which may involve the excessive release of pro-inflammatory factors and activation of immune system cells involved in regulating neurogenesis, synaptic plasticity, and neuronal survival process [8,9].

*Polygonatum sibiricmu* (PS), a traditional Chinese medicine commonly used in China, has the functions of nourishing the liver and kidneys and promoting longevity [10]. *Polygonatum sibiricmu* polysaccharides (PSP), as the main component of PS, have pharmacological effects such as immune regulation, preventing oxidative damage, and anti-inflammatory activity [11,12]; there are many studies showing that PSP has anti-aging capacity and mitigates memory dysfunction [13,14]. A typical concoction in traditional Chinese medicine, the “nine steaming and nine drying” method, is generally used to remove irritants from the drug, facilitate long-term storage, and enhance its biological activities [15,16]. Natural plant polysaccharides have structure-related antioxidant properties. For example, Li et al. have pointed out that the structural properties and antioxidant activity of PSP change after steaming [17]. Our previous studies have shown that PSP can ameliorate inflammation and relieve oxidative stress to prevent depression-like behaviors in mice [18,19]. However, the effect of steaming on the biochemical activity of PSP against memory impairment has not been studied, and its mechanism remains unclear.

Chronic administration of D-gal in rodents has been widely used to model brain aging [8,20]. In order to better understand the effect of the steaming process on the memory improvement effect of PSP, in this study, we isolated and purified PSPs with different steaming times and characterized the chemical compositions of PSPs. In addition, cellular experiments were explored to investigate the effect of the PSPs on antioxidant activity in HT-22 cells induced by D-gal. Animal experiments were explored in order to evaluate the effect of PSPs on D-gal-induced learning and memory dysfunction in mice. Furthermore, we assessed the effects of steamed PSPs on oxidation, inflammation, and synaptic function on the D-gal-induced aging process.

## 2. Results

### 2.1. Analysis of Physicochemical Properties of PSPs

With increased steaming time, the color of PS gradually changed from yellow-brown to dark, the aroma gradually became rich and thick, and the texture changed from slightly hard and tough to soft and sticky. The PSP solution gradually changed from a khaki color to a brown sugar color; with the increase in steaming time, the texture of freeze-dried PSPs changed from a powder structure (PSP0, PSP1, PSP2, PSP3) to a flocculent one (PSP4, PSP5, PSP6), then turned back to a powder (PSP7, PSP8, PSP9, PSP15). In addition, there were significant differences in the PSP contents before and after steaming. The recovery rates of PSP0, PSP1, PSP5, PSP9, and PSP15 were 35.95 ± 1.2%, 21.34 ± 0.7%, 15.75 ± 2.4%, 12.46 ± 1.8%, and 10.67 ± 1.5%, respectively. The polysaccharide content decreased with increased steaming time and then tended to be stable. This significant decrease in polysaccharides may be due to the massive hydrolysis of polysaccharides into monosaccharides and oligosaccharides under the continuous high temperature environment during the steaming process. In order to investigate whether the steaming process would cause changes in the physicochemical properties, we selected 0-steaming, 1-steaming, 5-steaming, 9-steaming and 15-steaming PSP samples to examine the physicochemical properties. Figure 1A shows the FT-IR spectra of PSP0, PSP1, PSP5, PSP9, and PSP15. The main absorption peaks of the FT-IR spectra of PSPs were around 3411, 2930, 1738, 1630, 1415, 1026, and 933 cm^−1^. The strong broad absorption peak around 3411 cm^−1^ was the intermolecular hydrogen bond O-H stretching vibration absorption peak, indicating the strong intra- and intermolecular interactions in the polysaccharide chains. The peaks around 2930 cm^−1^ were due to asymmetric and symmetric C-H bond stretching vibrations in aliphatic CH_2_, showing the typical absorption peaks of the polysaccharides. Glyoxylate characterized by carboxyl groups might cause three absorption peaks, with the absorption peak around 1738 cm^−1^ being due to the stretching vibration of the protonated C=O double bond and the absorption peaks around 1630 cm^−1^ and 1415 cm^−1^ to the protonated carboxylic acid COO-group. Therefore, PSP0, PSP1, PSP5, PSP9, and PSP15 are all acidic polysaccharides containing glyoxylate. The absorption peak around 1630 cm^−1^ became stronger with the increase in steaming time, indicating an increase in galacturonic acid after steaming. The strong absorption peak around 1026 cm^−1^ indicates that the monosaccharide was in the form of pyranoside. Below this, at 950 cm^−1^ there are multiple fingerprint characteristic peaks such as the stretching vibration of the sugar ring skeleton and the bending vibration of the fingerprint peak of the benzene ring.

Figure 1B shows the FT-IR spectra of PSPs with the second-order derivative process. In order to obtain richer information on the FT-IR absorption peaks, the average FT-IR spectra of the PSPs were subjected to the second-order derivative process and the resolution was improved by stripping the overlapping peaks. Among these, the intensity of PSP absorption peaks in the range of 1700–1500 cm^−1^ differed significantly, withPSP15 showing the strongest absorption and PSP1 showing the weakest absorption in this segment. PSP0 had the weakest absorption peak at 1630 cm^−1^, indicating the lowest calcium oxalate content, and PSP15 had the strongest absorption peak at 1630 cm^−1^, indicating the highest calcium oxalate content, i.e., the calcium oxalate content in PSPs increased with the increase in steaming time. All steamed PSPs had absorption peaks at 1415 cm^−1^, indicating methylene stretching vibrations, and all contained many long alkyl chains.

Characterization of the charge of polysaccharide indicated the presence of anionic groups in PSPs, which directly affects the stability of PSP-based solutions or colloids [21]. Therefore, we detected the zeta potential of PSPs (Figure 1C), which was about |−20 mV|; there was no significant difference among PSPs with different steaming times, indicating that the PSPs had a large negative charge and that stable PSP mixed solutions or colloids could be prepared [22]. Furthermore, pH results showed that all of the PSPs were weakly acidic, and that the acidity of the PSPs increased with the increase in steaming times (Figure 1E).

XRD analysis is an important method used to analyze the structure of polymers and compounds [23]; the crystalline state of polysaccharides determines physical properties such as their flexibility and solubility [24]. With the XRD pattern smoothed by MDJ jade5.0 software, it can be clearly observed that the diffraction peaks and intensities of different PSPs in the 10~90° range of the diffraction angle are slightly different (Figure 1D). These results indicate that the crystallization state of the PSPs was essentially the same and that the steaming process did not destroy the crystalline state of the PSPs. As shown in Figure 1D, 2θ did not have strong diffraction absorption peaks in the range of 10~90°; only a very few small diffraction peaks existed, with a large bun diffraction peak appearing around the diffraction angle of 20°, which indicates that the PSPs were amorphous polymers, not a polymer with submicrocrystalline and amorphous coexistence. This property could confirm the purity of PSPs.

### 2.2. Steamed PSPs Promoted Antioxidant Capacity in D-Gal-Treated HT-22 Cells

As shown in Figure 2, the results of our flow cytometry analysis showed that the fluorescence intensity of DCFH-DA in D-gal-treated HT-22 cells was significantly higher than that in the control group. Compared with the model group, the fluorescence intensity of DCFH-DA was significantly lower in the steamed PSP groups (*p* < 0.05, F _(6, 14)_ = 402.8). After D-gal treatment, the cytoplasmic reactive oxygen species (ROS) production of HT-22 cells increased significantly, and steamed PSPs inhibited the accumulation of ROS in D-gal-induced HT-22 cells, suggesting that steamed PSPs had strong antioxidant capacity.

### 2.3. Steamed PSPs Improved Spleen Index and Behavioral Functions in D-Gal-Injured Mice

During the administration of D-gal and PSPs, we examined the changes in body weight for eight consecutive weeks; no significant differences were observed in the body weight of the five groups (*p* > 0.05, Figure 3A). After the behavioral tests, we collected the spleens of the mice. As shown in the Figure 3B, the spleen index of mice in the D-gal group was significantly lower than that in the control group. By contrast, PSP treatment could improve the spleen index of mice injured by D-gal. Among them, PSP5 showed a better effect in comparison with PSP0 (*p* < 0.05, F _(4, 35)_ = 15.64).

Twenty-four hours after the last administration of D-gal and PSPs, we performed behavioral tests. As shown in Figure 3C, there was no difference in the total motor distance in OFT among the groups, indicating that D-gal administration did not impair motor function of the mice (*p* > 0.05, F _(4, 35)_ = 1.273). In the OFT, mice in the D-gal group showed a significant decrease in central motor distance compared to mice in the control group, while PSP administration significantly increased central motor distance (0, 5, and 9 steamings) compared with D-gal group; a significant increase was found in the PSP5 and PSP9 groups compared to the PSP0 group (*p* < 0.05, F _(4, 35)_ = 29.64, Figure 3D). These results suggest that the steamed PSPs were able to ameliorate the D-gal-induced anxiety-like behaviors, with the improvement effect of PSP5 and PSP9 being more pronounced.

In order to assess the effect of PSPs on the spatial learning and memory function of D-gal-injured mice, we conducted a five-day navigation experiment and a one-day spatial exploration experiment. Figure 3E shows that in the navigation experiment, the escape latency was significantly increased in the D-gal group compared to the control group and significantly decreased in the PSP5 group compared to the D-gal group. The mice in PSP5 group had shorter escape latency than mice in the PSP0 group on the second and fourth days. Figure 3F (*p* < 0.05, F _(4, 35)_ = 8.362) and Figure 3G (*p* < 0.05, F _(4, 35)_ = 3.637) shows the results of withdrawal platform spatial exploration experiment on the sixth day. Compared with control group, both the number of platform crossings and the dwelling time in the platform quadrant were significantly reduced in the D-gal group, while the number of platform crossings and dwelling time in the platform quadrant were significantly increased in the PSP5 group compared to the D-gal group. These results suggest that PSP5 can effectively improve D-gal-induced learning and memory impairment in mice.

### 2.4. Steamed PSPs Prevented D-Gal-Induced Morphological Changes of Hippocampal Pyramidal Cells 

The structure of hippocampal neurons was detected by HE staining. In the control group, the pyramidal cells in the CA1, CA3, and DG regions were round and intact with clearly visible nuclei. By contrast, in the D-gal group the degenerated pyramidal cells showed deeper staining (nuclear fixation) with blurred nucleoplasmic borders, the number of normal pyramidal cells in the three regions was significantly reduced, and the number of degenerated cells was significantly increased (Figure 4A). PSP5 effectively prevented the degeneration of pyramidal cells in CA1 area in D-gal-injured mice (*p* < 0.05, F _(4, 35)_ = 16.00, Figure 4B; *p* < 0.05, F _(4, 35)_ = 18.24, Figure 4E). Meanwhile, PSP5 and PSP9 effectively prevented the degeneration of pyramidal cells in the DG area (*p* < 0.05, F _(4, 35)_ = 42.26, Figure 4C; *p* < 0.05, F _(4, 35)_ = 12.16, Figure 4F). PSP0, PSP5, and PSP9 effectively prevented the degeneration of pyramidal cells in the CA3 region, with PSP5 the most effective (*p* < 0.05, F _(4, 35)_ = 27.96, Figure 4D; *p* < 0.05, F _(4, 35)_ = 21.91, Figure 4G). The above results suggest that PSP treatment can prevent D-gal-injured pathological changes in hippocampal neurons, with the treatment effect of PSP5 being the best.

### 2.5. Effects of Different Steamed PSPs on HO-1/Nrf2 Signaling Pathway in D-Gal-Induced Mice

In our examination of the markers of oxidative damage in serum, the MDA level was significantly increased in D-gal-injured mice compared with the control group, while PSP0 and PSP5 significantly reversed those changes. In addition, the MDA level of the PSP5 group was significantly decreased compared with the PSP9 group (*p* < 0.05, F_(4, 35)_ = 11.97, Figure 5A). The results of Western blotting showed that expression of both HO-1 and Nrf2 in the D-gal group was significantly decreased compared with control group, indicating that the level of oxidative stress in the mice increased significantly after D-gal injection. Compared with the D-gal group, the levels of HO-1 and Nrf2 were significantly increased after PSP9 administration. Additionally, the level of HO-1 was significantly increased in the PSP9 group compared with the PSP0 group (*p* < 0.05, F _(4, 25)_ = 5.849, Figure 5B; *p* < 0.05, F _(4, 25)_ = 6.456, Figure 5C). These results suggest that the steaming process can promote the anti-oxidative capacity of PSP in D-gal-injured mice.

### 2.6. Steamed PSPs Prevented D-Gal-Induced Neuroinflammation in Mice

We investigated the effect of PSPs on inflammatory factors in the serum of D-gal-injured mice. Figure 6A,B showed that 60 days after D-gal injection, the levels of IL-1β and TNF-α in the D-gal group were significantly higher than those in the control group, and that PSPs significantly reduced IL-1β and TNF-α (*p* < 0.05, F _(4, 25)_ = 14.94, F _(4, 35)_ = 11.95). Furthermore, the results of Western blotting showed that D-gal significantly increased the expression of NLRP3, which was prevented by the PSPs (*p* < 0.05, F _(4, 25)_ = 8.441, Figure 6D). There was an upward trend in ASC in the D-gal group, which was mitigated by PSP5 (*p* < 0.05, F _(4, 25)_ = 3.812, Figure 6C).

### 2.7. Steamed PSPs Improved D-Gal-Induced Synapse Deficits in the Hippocampus

We found that expression of GluA1, GluA2, GluN2A, and GluN2B in the hippocampus was significantly decreased in the D-gal group, and that both PSP0 and PSP5 could prevent the down-regulation of those proteins induced by D-gal. Compared with PSP9, PSP5 significantly increased GluA1 expression. Compared with PSP0 and PSP9, PSP5 significantly increased GluA2 expression. Compared with PSP9, PSP5 significantly increased the expression of GluN2A. Compared with PSP0 and PSP9, PSP5 significantly increased GluN2B expression (*p* < 0.05, F _(4, 25)_ = 10.32, F _(4, 25)_ = 7.201, F _(4, 25)_ = 5.387, F _(4, 25)_ = 29.78, F _(4, 20)_ = 1.700, Figure 7A–D).

PSD95 and Arc are proteins closely related to synaptic plasticity [25]. Compared with the control group, the levels of PSD95 and Arc in the D-gal group were significantly decreased, while PSP0 and PSP5 administration significantly prevented the decrease in Arc level, and Arc in the PSP0 and PSP5 groups was higher than that in PSP9 group (*p* < 0.05, F _(4, 25)_ = 13.40, Figure 7E). In addition, after administration of PSP5 and PSP9, the decrease in the level of PSD95 was significantly prevented, and PSD95 level was higher in the PSP9 group compared with that in PSP0 group (*p* < 0.05, F _(4, 25)_ = 8.592, Figure 7F).

Transmission electron microscope (TEM) measurement showed that after D-gal injection the number of synapses in the hippocampus was significantly reduced compared with control. This was prevented by PSP5 and PSP9 treatment, and the PSD95 level in the PSP9 group was higher than that in the PSP0 group (*p* < 0.05, F _(4, 20)_ = 15.18, Figure 7G,H). These results suggest that PSPs can protect against D-gal-induced synaptic damage in the hippocampus.

## 3. Discussion

In this study, we investigated the effect of steamed PSPs on memory impairment in D-gal-induced mice. To the best of our knowledge, this is the first time steamed PSPs have been used as therapeutic agents in D-gal-treated mice. We demonstrated that the steaming process alters the physicochemical properties and pharmacological effects of PSPs and that steamed PSPs likely improve memory impairment by alleviating oxidative stress, neuroinflammation, and synaptic damage. This study points to a strategy for using steamed PSPs in multi-targeted treatment of aging-related diseases, and can provide insight for drug development based on steamed PSPs.

### 3.1. The Steaming Process Enhances the Biological Activity of PSP

The steaming process is able to increase pharmacological activity of PSP while reducing toxicity and enhancing flavor. The Maillard reaction is involved in the steaming process of a variety of herbal medicines [26]. Maillard reaction products have various properties, such as antioxidant activity, chemopreventive activity, and antimutagenic activity [27]. In our study, the appearance of PSs changed after steaming; the texture became tougher, the surface was more compact, the color deepened, and the aroma gradually became more rich and thick, all of which might be caused by the Maillard reaction. Interestingly, previous research has shown a gradual increase in the antioxidant capacity of PSPs with increased steaming time [28]. Our study demonstrated that steamed PSPs express strong antioxidant capacity by preventing the production of ROS in D-gal-induced HT-22 cells. However, our flow cytometry results showed that the antioxidant capacity of PSP1 was slightly lower than that of PSP0, which may be due to the fact that the structural properties of PSPs did not change much at the initial stage of steaming, while the antioxidant activity fluctuated slightly. The above-mentioned research reports pointed out that the surface of PSPs becomes more compact with steaming, probably due to the degradation and aggregation effects of the process. Therefore, we characterized the structural features of PSPs by FT-IR analysis; the results show that the spectra of PSPs had similar characteristic absorption before and after steaming, indicating that the structure of PSPs prepared before and after steaming is similar. However, the content of glyoxalate increased and became more acidic with the increase of steaming times, which is consistent with the results when measuring the pH value. The crystalline structure of PSPs directly affects the physical properties, including the solubility, swelling, viscosity, and opacity [21]. The steaming process did not alter the crystalline structure of the PSPs in our XRD studies. The zeta potential is the potential difference between the continuous phase and the fluid stabilized layer attached to the dispersed particles. It is significant because the value is related to the stability of the colloidal dispersion and is a measure of the strength of mutual repulsion or attraction between the particles. In general, the higher the absolute value of the zeta potential, the more stable the system is. The results of our study showed that the zeta potential of steamed PSPs was around |−20 mV|. The chemical composition and structure of PSPs were the main factors controlling their pharmacological efficacy, and the above studies on the physicochemical properties of steamed PSP have shown that the steaming process changes the structural properties of PSPs and improves their antioxidant capacity and pharmacological activity. However, in the Qing Dynasty “Yao Pin Bian Yi” of He Shou Wu, it is written: “Long steamed and cooked, it becomes purple-black”. The “long” here represents the length of the steaming time period or the times, although the number has not yet been determined. Therefore, research on the “nine steaming and nine drying” method of PS concoction needs further examination.

### 3.2. Steamed PSPs Ameliorate Behavioral Impairment in D-Gal-Induced Mice

The rhizomes of PS are often used as tonic and functional foods, and are marketed by repeated steaming and drying to modify their taste and increase their tonic effect. As one of the most important constituents of PS, PSP has various biological activities, including anti-tumor, antioxidant, anti-inflammatory, and immunomodulatory, and plays an important role in the treatment of various diseases [29,30]. There are many reports that PSP improves memory dysfunction in mice [31,32], although the effect of PSP on memory impairment after steaming has not been extensively studied and its mechanism is unclear. In our study, the physicochemical properties of PSPs were altered to varying degrees after steaming, and we used OFT and MWM to assess the ameliorative effects of the changes in the physicochemical properties of steamed PSPs on the impaired spontaneous motor function and learning memory function in D-gal-induced mice. The results show that the steaming process can increase the biological activity of PSPs, thereby improving behavioral impairment in D-gal-induced mice; this may be due to the antioxidant activity of PSPs as well as the alteration of chemical structure during the concoction process. PSP5 showed optimal improvement in memory impairment in D-gal-induced mice, which may suggest that an increase in steaming time may not be necessary for better efficacy of PSPs. Therefore, choosing a suitable steaming time, such as five, may be more effective, thus avoiding the waste of resources in the production of “nine steaming and nine drying” varieties. In this study, we found that PSP5 could improve memory dysfunction better relative to either PSP0 or PSP9, however, it is not certain that five-steaming is the optimal steaming time to exert the medicinal effect, and it is worthwhile to investigate more deeply whether and how much the effect of steamed PSPs can be improved in other models by selecting a more appropriate steaming methods and times.

### 3.3. Steamed PSPs Have Protective Effects against D-Gal-Induced Pyramidal Cell Injury in the Hippocampus

The hippocampus is divided into many anatomically distinct but closely related regions, and the major cellular regions of the hippocampus are sensitive to aging [33]. It has been suggested that CA3 neurons of the hippocampus act as an auto-associative network to form and store declarative memories, and that these memories are further recoded in the CA1 [34]. In addition, memory for specific locations may be affected by the degree of overlap of key distal spatial cues, while the dentate gyrus (DG) of the hippocampus plays an important role in the acquisition of new information; DG-injured rats show deficits in spatial memory [35]. In this study, we evaluated the pathological changes in these regions of the hippocampus. The results of HE staining showed that the CA1, CA3, and DG neurons in the hippocampus of the D-gal group mice were significantly damaged, and that as the pathological grade increased, the steamed PSPs could alleviate this pathological change. Interestingly, the effect of PSP5 was more obvious than that of PSP0 or PSP9, which may suggest that the effect of PSP5 in improving learning and memory impairment might be achieved by improving damage to hippocampal neurons. Therefore, the ameliorating effect of steamed PSPs on learning and memory dysfunction in D-gal-induced mice might be directly correlated with the protective effect on neurons in the CA1, CA3, and DG regions of the hippocampus.

### 3.4. Steamed PSPs Improve Memory Dysfunction in D-Gal-Induced Mice by Enhancing Antioxidant Capacity

A mild level of oxidative stress in the nervous system is important for maintaining physiological function, and imbalances of oxidative stress lead to neurological dysfunction. Nrf2 is a ubiquitous transcriptional activator that attenuates oxidative stress in neurodegenerative diseases by promoting phase II antioxidant enzymes, including HO-1 [36]. The Nrf2/HO-1 axis is the main pathway and primary sensor against external oxidative stimuli [37]. Neuroinflammation is a pathological feature of neurodegenerative diseases, especially in the presence of over-activated and dysregulated microglia; these secrete cytotoxic reactive oxygen and nitrogen species, with deleterious effects on neurons [38]. Nrf2 is a common therapeutic target in neurodegenerative diseases and can reduce NLRP3 overactivation, thereby reducing NLRP3-mediated neuroinflammation [39]. MDA is produced by the interaction of oxygen radicals and lipids. It is cytotoxic and can cross-link proteins and nucleic acids, thereby inactivating proteins, leading to cellular dysfunction and reflecting the extent of damage caused by ROS. By studying galactokinase-deficient GM00334 fibroblasts, it has been found that excessive D-gal metabolism generates large amounts of ROS in the cells, leading to oxidative stress damage and inducing cellular damage [40]. In this study, we found that MDA level was significantly elevated, HO-1 and Nrf2 protein expression were significantly decreased, and NLRP3 levels were significantly elevated in the hippocampus of mice in the D-gal group. Steamed PSPs were able to significantly reverse this alteration, suggesting that the improvement of memory dysfunction in D-gal-induced mice by steamed PSPs might be achieved through its antioxidant and anti-inflammatory effects.

NLRP3 inflammasome signaling can trigger the production of the pro-inflammatory mediator IL-1β, which in turn releases the pro-inflammatory cytokine TNF-α [41]. In addition, oxidative damage induces neuroinflammation, as evidenced in our study by increased production of pro-inflammatory factors such as IL-1β and TNF-α. Furthermore, oxidative stress and neuroinflammation promote each other, and together contribute to the progression of memory impairment and neuropathology [42]. However, our results showed no difference in the reversal of neuroinflammation among PSPs with different steaming times, indicating that the differential efficacy of steamed PSPs may be mediated through modulation of antioxidant stress activity rather than through neuroinflammatory pathways.

### 3.5. Steamed PSPs Prevent Synaptic Deficits in D-Gal-Induced Mice

Aging is a physiological process that leads to behavioral changes associated with changes in structural, neurochemical, and physiological processes in the brain [43]. High levels of oxidative stress induce a state of senescence in cells, with major pathological features including DNA damage, alterations in the cell cycle, and telomere damage [44]. In later stages of life, it is associated with cognitive decline, synaptic loss, and the development of neuronal degeneration [45]. Synaptic deficit leads to cognitive impairment and aging-related brain disorders. We examined alterations in protein such as AMPAR, NMDAR, PSD95, and Arc, which are closely related to synaptic plasticity; the results show that D-gal significantly reduces the expression of these proteins, while steamed PSPs were able to reversed the changes. In addition, PSP5 was more effective in reversing the decrease in AMPAR, NMDAR, and Arc, which may reveal that PSP5 is more effective in treating synaptic deficit induced by D-gal. This might explain the better effect of PSP5 in reversing behavioral changes in our OFT and MWM tests.

Our study has limitations that should be taken into considerations in future research. First, we did not quantify the composition ratio of different monosaccharides in steamed PSPs. As reported, after the steaming process, the types of monosaccharides were similar, but their ratios presented obvious differences. Compared with PSP0, the level of mannose decreased gradually, while the levels of glucose and arabinose increased during the steaming process. The level of galactose first decreased and then increased after the fourth steaming [17]. These changes in monosaccharide content after steaming might explain the enhanced biological activity. Additionally, in a previous study we analyzed the monosaccharide composition of non-steamed PSP by UHPLC [18], and our findings were consistent with the literature. Therefore, we did not repeat this part of experiment. Second, aging is the result of multiple mechanisms. In this study, we applied D-gal as an aging inducer. Whether steamed PSPs have an ameliorative effect on memory impairment caused by natural aging remains an open question. Finally, we investigated the preventive effect of PSPs on memory impairment during the aging process, not the therapeutic effect of PSPs on memory impairment caused by aging. Therefore, the therapeutic effect of PSPs against aging-related memory impairment should be further investigated in future studies.

In conclusion, the steaming process alters the physicochemical properties of PSPs and promotes their antioxidant capacity. Steamed PSPs can ameliorate the impairment of learning and memory induced by D-gal, likely by preventing oxidative stress and reducing inflammation, which in turn protect synaptic plasticity and neuronal damage through the Nrf2/HO-1 signaling pathway. This study provides a potential molecular mechanism for the application of steamed PSPs to prevent aging-related memory dysfunction.

## 4. Materials and Methods

### 4.1. Animals and Drugs

Male C57BL/6 mice (3 months old, body weight 20–25 g) were purchased from the Experimental Animal Center of Anhui Medical University (SCXK (Anhui) 201-002). Mice were housed at 22 ± 2 °C, 45–65% humidity, and underwent a 12/12-h light/dark cycle with ad libitum access to food and water. After the behavioral experiments, the animals were decapitated under anesthesia with isoflurane. The experimental protocol was supervised by the Ethics Committee of Anhui University of Chinese Medicine (Approval No. AHUCMmouse-20210205, 5 February 2021).

### 4.2. Preparation of PSPs

After removing impurities and fibrous roots, fresh PS was washed, dried, and then divided into eleven sections and marked with PS0, PS1~PS9, and PS15. The PSs were weighed and recorded, then PS0 was put directly into the oven at 55 °C and dried until the weight did not decrease. The remaining ten samples were steamed above water for 2 h, then taken out and dried in an oven at 55 °C until the weight did not change. PS1 was removed and the remaining nine samples were steamed above water for 2 h. The same operation was repeated several times to obtain PS0~PS9 and PS15, respectively. Subsequently, the PSs were polished into powder, and 80 g PSs were soaked in ultrapure water for 2 h then refluxed with double distilled water for 2 h each time. Afterwards, the aqueous extract of PS was concentrated by rotary evaporation, and eight times the volume of absolute ethanol was added at 4 °C. The crude polysaccharide was obtained by centrifugation and drying after precipitation. Subsequently, the crude polysaccharide was dissolved in ultrapure water. Sevag reagent was used for 5 ~ 6 extractions, which comprised chloroform and n-butyl alcohol at a volume ratio of 4:1. We then detected the samples, and UV spectrophotometer analysis revealed no absorbance peak at 280 nm, suggesting an absence of protein in the PSP solution. The residual organic solvent was removed with a rotary evaporator. After freeze-drying, refined PSPs were obtained, which were named PSP0~PSP9 and PSP15, and the PSPs were weighed and recorded.

### 4.3. Determination of Physicochemical Properties of PSPs

#### 4.3.1. Fourier Transform Infrared Spectrometer (FT-IR)

PSPs were passed through an 80-mesh sieve, 1 mg of the sample powder was put into an agate mortar, 200 mg of potassium bromide powder was added as a dispersant, and the powder was ground evenly. The spectral scanning range was 4000–500 cm^−1^, and the resolution was 4 cm^−1^ for 32 scans. The background of H_2_O and CO_2_ was deducted in real time during scanning. The spectral data were processed by OMNIC 8.0 software for coordinate normalization and baseline correction.

#### 4.3.2. Zeta Potential Test

The Zeta potential of the steamed PSP solutions was measured using a Malvern Nano-zs nanoparticle analyzer and particle sizer. All samples were diluted to concentrations of 5 × 10^−3^ mg/mL and 5 × 10^−4^ mg/mL, and the measurements were repeated three times.

#### 4.3.3. pH Value Measurement

The pH value of the PSP solutions was measured using a pH meter; the measurement was repeated three times. The sample concentration was 100 mg/mL.

#### 4.3.4. X-ray Diffraction (XRD) Analysis

An X-ray polycrystalline powder diffractometer (model: Philips X’pert X-ray diffractometer, Cu Kα, excitation wavelength λ = 1.54182 Å) was used to qualitatively analyze the phases of the PSPs with different steaming times and observe any similarities or differences in their crystallization.

### 4.4. Cell Culture and Oxidative Capacity Assay

HT-22 cells were cultured in DMEM (high glucose) containing 10% fetal bovine serum (Gibco, NY, USA) in a 37 °C, 5% CO_2_ incubator. The medium contained 100 U/L penicillin and 100 ug/L streptomycin. When confluence reached 80–90%, the cells were washed twice with PBS and digested with 0.25% trypsin for about 1 min to prepare a single cell suspension. After passaging, the cells were seeded into culture flasks or plates for experiments.

The HT-22 cells in the logarithmic growth phase were taken, digested with trypsin, and inoculated into six-well plates. A control group, model group, and multiple PSP treatment groups were set up. The control group was added with 10% FBS-containing medium and the model group was added with 20 mg/mL D-gal and 10% FBS-containing medium [46]. The PSP treatment groups were pre-treated with 200 μg/mL PSP0, PSP1, PSP5, PSP9, and PSP15 for 2 h, respectively, then 20 mg/mL D-gal was added and co-cultured for 24 h. Cells from these groups were collected and incubated in a 37 °C incubator for 20 min with DCFH-DA at a final concentration of 10 μM. After washing with serum-free medium three times, the cells were tested by flow cytometry (Beckman Coulter, Inc.) and the relative fluorescence intensity was analyzed by flowJo v.10 software.

### 4.5. Animal Model and Treatments

The mice were divided into five groups, namely, a control group, D-gal group, D-gal + PSP0 group, D-gal + PSP5 group, and D-gal + PSP9 group, with eight mice per group. After one week of adaptation, the model was established by subcutaneous injection of D-gal. The mice in the control group received the same volume of saline (i.sc). From the first day during preparation of the model, the mice in the D-gal + PSP groups were respectively administered PSP0, PSP5, and PSP9 by gavage for 60 days at a dose of 400 mg/kg [19], with the PSPs dissolved in ultrapure water. The control group and the D-gal group were administered water by gavage for 60 days. Except for the control group, the groups were subcutaneously injected D-gal for 60 days at a dose of 400 mg/kg. Body weight was measured once a week. After 60 days, behavioral tests were performed and hippocampus, spleen, and serum were immediately collected after behavioral tests for further analysis. Finally, eight mice per group were included in the behavioral tests, and body weight tests, and measurement of spleen index. Bilateral hippocampi of four mice in each were used in the pathological analysis and the other four mice in each group were used in Western blotting.

### 4.6. Morris Water Maze (MWM)

The MWM test included a five-day positioning navigation experiment and a one-day platform withdrawal space exploration experiment. Prior to the experiment the mice received adaptive training for two days, i.e., they were allowed to swim freely for 60 s to grow familiar with the swimming pool environment and the grasping and other operations of the experiments. The first fifth days consisted of the positioning navigation experiment. The mice were placed into the water starting from the 1st, 2nd, 3rd, and 4th quadrants facing the pool wall and the total time for the mice to find the hidden platform within 60 s was recorded, with times longer than 60 s recorded as 60 s. On the sixth day, the space exploration experiment began with the withdrawal of the platform. The test time for each mouse was 60 s, and the swimming time and the number of crossing the platform were recorded in the platform quadrant.

### 4.7. Open Field Test (OFT)

The OFT was used to evaluate the autonomous exploration behavior of the animals. The OFT was carried out in a quiet environment; the mice were placed in the bottom center of the box for 5 min. After each test, it was necessary to clean the inner wall and bottom of the box with 75% alcohol. The total distance and center distance were recorded for comparison among the groups.

### 4.8. Hematoxylin and Eosin Staining (HE Staining)

After the behavioral tests, the whole brains of the mice were perfused with PBS, fixed with 4% paraformaldehyde, embedded in paraffin, and sectioned (5-μm thickness), followed by dewaxing, dyeing, and sealing. The hippocampal neurons were detected by HE staining. The numbers of normal pyramidal cells and degenerated cells in the DG, CA1, and CA3 regions of the hippocampi (bregma: −2.4 mm ± 10 μm) were counted using ImageJ software.

### 4.9. Spleen Index

After the behavioral test, the mice were weighed. The spleens were taken for weighing and the spleen indexes were calculated by the following formula: spleen index = spleen mass (mg)/mouse body mass (g).

### 4.10. Transmission Electron Microscope (TEM)

The hippocampi were quickly separated on ice and fixed in 2.5% glutaraldehyde (pH = 7.4), dehydrated, permeabilized, sectioned, and double-stained with 4% uranium acetate and 0.5% structural lead citrate. Transmission electron microscopy was used to observe the changes in the ultrastructure of neurons in the hippocampal CA1 area.

### 4.11. Determination of Malondialdehyde (MDA) and Cytokine Detection

The level of MDA in the serum from the mice was detected using an MDA detection kit (BC0175, Solarbio). Then, the serum was placed on ice and the assay was performed according to the manufacturer′s instructions. The absorbance at the corresponding wavelength was measured and the MDA content was calculated according to the standard curve.

The levels of TNF-α and IL-1β in the serum were detected by ELISA. After centrifugation (1000× *g*, 20 min), the supernatant was collected for protein quantification. Absorbance was quickly read on a microplate reader at a detection wavelength of 450 nm. The concentrations of IL-1β and TNF-α were calculated according to the standard curve.

### 4.12. Western Blotting

The hippocampi were grounded in lysis buffer (RIPA) for 30 min on ice, then centrifuged at 12,000 rpm for 5 min at 4 °C. Thereafter, protein samples were separated by 10% sodium dodecyl sulfate-polyacrylamide gel electrophoresis at 75 V for 15 min, followed by 115 V for 75 min before transferring at 300 mA onto a nitrocellulose membrane for 2 h. Membranes were blocked with 5% skim milk (dissolved in PBS containing 0.1% Tween-20) for 2 h at room temperature, washed three times with PBS for 10 min each, then incubated at 4 °C overnight in primary antibody against the α-amino-3-hydroxy-5-methyl-4-isoxazole-propionic acid receptor (AMPAR) subunits GluA1 (1:1000, CST) and GluA2 (1:1000, CST), the N-methyl-D-aspartate receptor (NMDAR) subunits GluN2A (1:1000, CST), GluN2B (1:1000, CST), and p-GluN2B (1:1000, CST), nuclear factor-erythroid factor 2-related factor 2 (Nrf2, 1:1000, ZEN BIO), heme oxygenase-1 (HO-1, 1:1000, ZEN BIO), NOD-like receptor protein 3 (NLRP3, 1:1000, CST), apoptosis-associated speck like protein (ASC, 1:1000, ZEN BIO), postsynaptic density protein 95 (PSD95, 1:1000, CST), activity-regulated cytoskeletal protein (Arc, 1:1000, Synaptic Systems), and β-actin (1:1000, CST). The following day, the samples were incubated in peroxidase-conjugated goat anti-rabbit IgG (1:10,000; Zs-Bio, Beijing, China) for 2 h at room temperature. Densitometric quantification of band intensities was performed using ImageJ v.10 software (National Institutes of Health). Quantification of the target proteins was determined relative to β-actin, and levels of phosphorylated protein were determined relative to total protein.

### 4.13. Data Analysis

Data were analyzed using GraphPad Prism 8.0 software (GraphPad Inc., San Diego, CA, USA) and presented as mean ± SEM values; *p* < 0.05 was considered significantly different. Differences between groups were compared using one-way ANOVA followed by Bonferroni test.

## Figures and Tables

**Figure 1 ijms-23-11220-f001:**
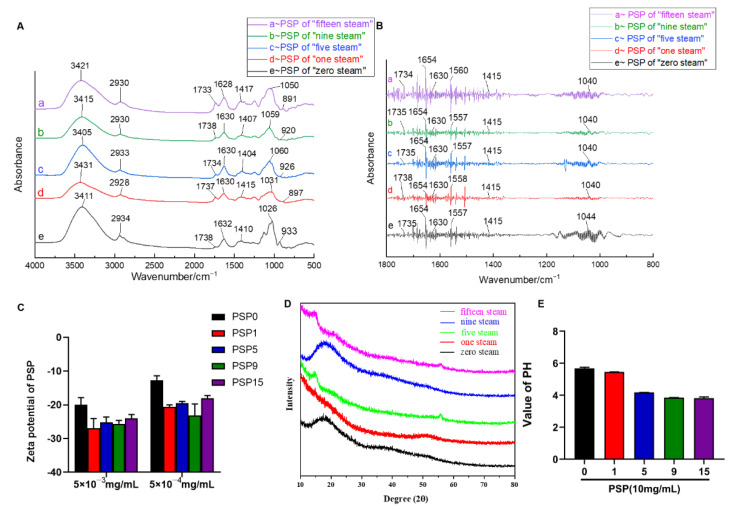
Physiochemical analysis of PSPs with five steaming times. (**A**) FT-IR spectra analysis of PSPs; (**B**) second derivative infrared spectrum of PSPs; (**C**) zeta potential of PSPs; (**D**) particle X-ray diffraction curves of PSPs; (**E**) pH value of PSPs.

**Figure 2 ijms-23-11220-f002:**
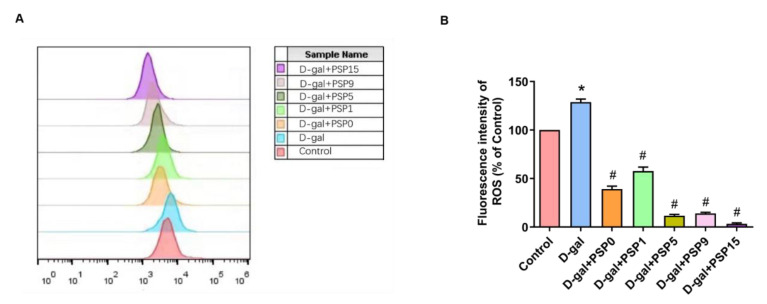
Steamed PSPs increase the antioxidant capacity of D-gal-induced HT-22 cells. (**A**) Fluorescence intensity of ROS; (**B**) effects of PSP on ROS of HT-22 cells (hippocampal neuron of mice). Data are expressed as means ± SEM. * *p* < 0.05 vs. Control, ^#^ *p* < 0.05 vs. D-gal.

**Figure 3 ijms-23-11220-f003:**
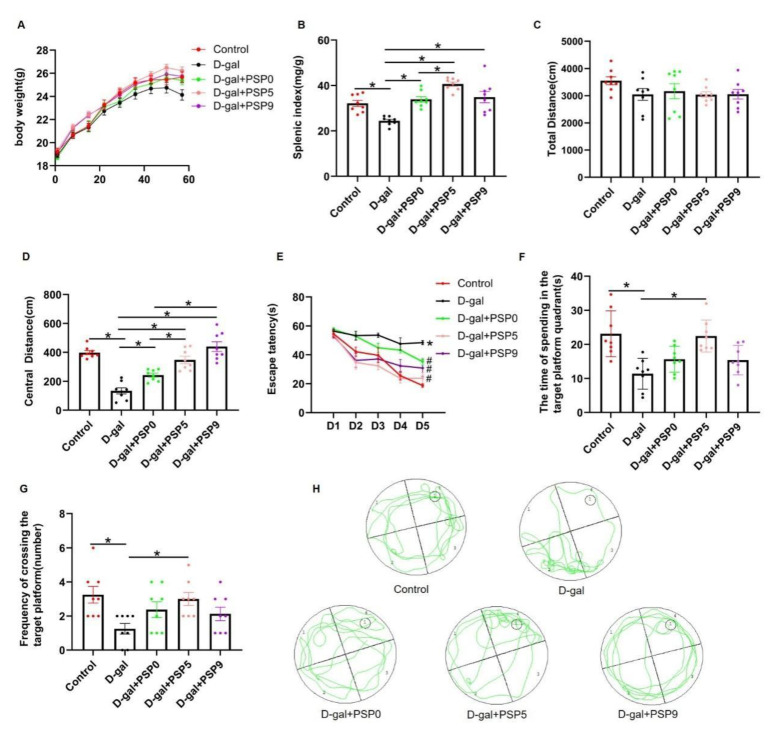
Steamed PSPs ameliorate the decrease of splenic index and behavioral defects in D-gal-induced mice. Steamed PSPs increased (**A**) body weight and (**B**) spleen index in D-gal-injured mice. (**C**,**D**) Total distance and central distance traveled in the OFT (*n* = 8 per group). (**E**) Escape latency, (**F**) time spent in the target platform quadrant, and (**G**) frequency of crossing the target platform in the MWM; (**H**) traces of the MWM. Data are expressed as means ± SEM (*n* = 8 per group). * *p* < 0.05 vs. Control, ^#^ *p* < 0.05 vs. D-gal.

**Figure 4 ijms-23-11220-f004:**
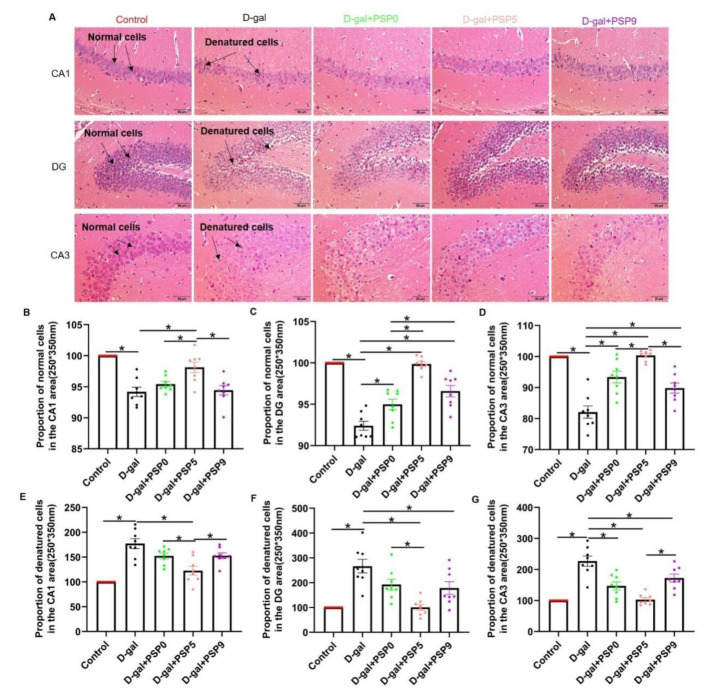
Steamed PSPs prevent the loss and degeneration of neurons in the hippocampal CA1, DG, and CA3 regions. (**A**) Hematoxylin and eosin (H&E) staining in the hippocampal regions CA1, CA3, and DG (normal cells are round and intact with clearly visible nuclei; in contrast, degenerated cells are more heavily stained (pyknosis) and have blurred nucleocytoplasmic borders) (Magnification: x 200). (**B**–**G**) Normal cells and denatured cells in the hippocampal CA1, DG, and CA3 region (*n* = 8). Data are expressed as means ± SEM. * *p* < 0.05.

**Figure 5 ijms-23-11220-f005:**
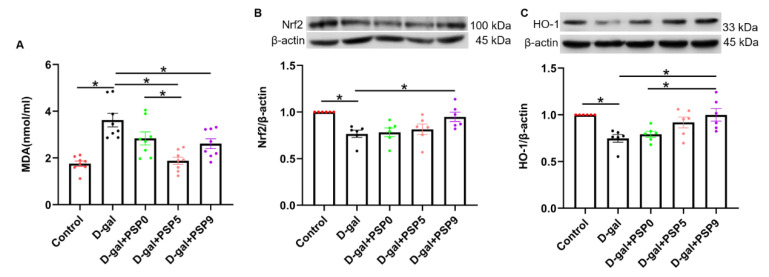
Steamed PSPs attenuate D-gal-induced oxidative damage by regulating the HO-1/Nrf2 signaling pathway. (**A**) MDA level in the serum (*n* = 8). (**B**,**C**) The upper panels show representative blots of Nrf2/β-actin and HO-1/β-actin, while the lower panel shows Nrf2/β-actin and HO-1/β-actin quantification data (*n* = 6). Data are expressed as means ± SEM. * *p* < 0.05.

**Figure 6 ijms-23-11220-f006:**
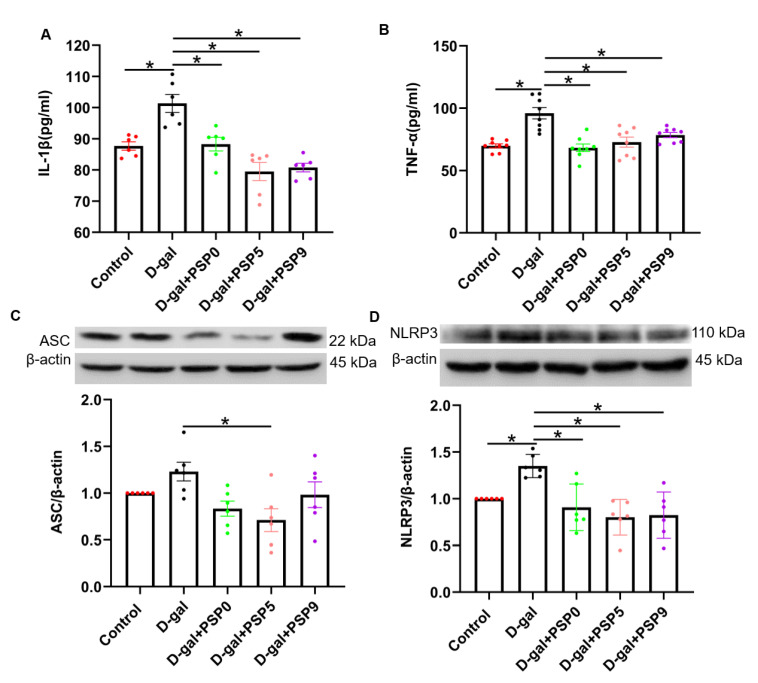
Steamed PSPs inhibit D-gal-induced neuroinflammation. (**A**,**B**) TNF-α and IL-1β levels in the serum (*n* = 6–8). (**C**,**D**) The upper panels show representative blots of ASC/β-actin and NLRP3/β-actin, while the lower panel shows ASC/β-actin and NLRP3/β-actin quantification data (*n* = 6). Data are expressed as means ± SEM. * *p* < 0.05.

**Figure 7 ijms-23-11220-f007:**
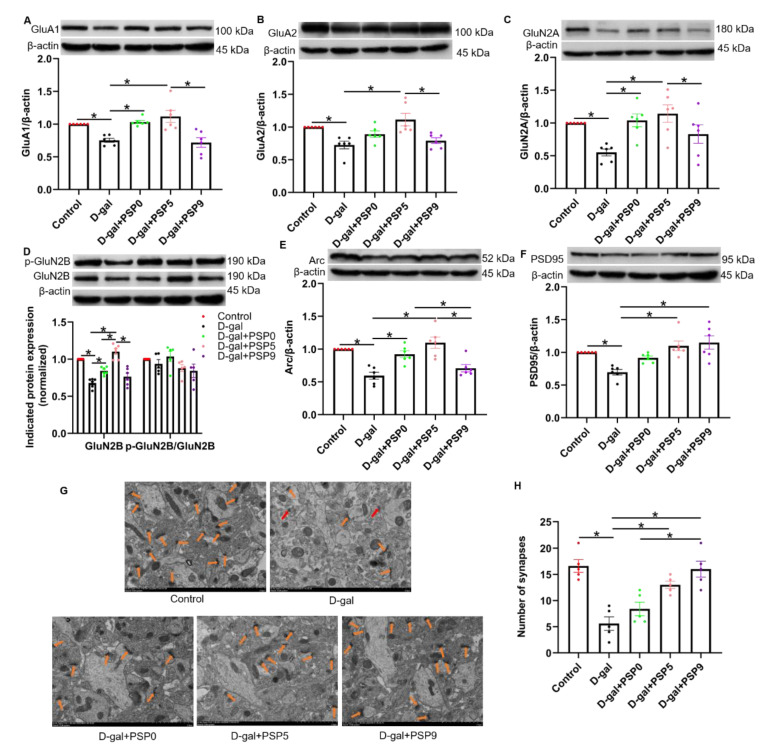
Steamed PSPs ameliorate D-gal-induced hippocampal synaptic deficits. (**A**–**F**) The upper panels show representative blots of Arc/β-actin, GluA1/β-actin, GluA2/β-actin, GluN2A/β-actin, GluN2B/β-actin, p-GluN2B/GluN2B, and PSD95/β-actin, while the lower panel shows Arc/β-actin, GluA1/β-actin, GluA2/β-actin, GluN2A/β-actin, GluN2B/β-actin, p-GluN2B/GluN2B, and PSD95/β-actin quantification data (*n* = 6). (**G**) Representative images of synaptic structure in hippocampal area CA1 (Magnification: ×5000). Red arrows indicate the synapses. (**H**) Number of synapses (12 μm^2^) (*n* = 5). Data are expressed as means ± SEM. * *p* < 0.05.

## Data Availability

The data that support the findings of this study are available from the corresponding authors, G.Z. and X.W., upon reasonable request.

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
