# Peer review of "Use of Steaming Process to Improve Biochemical Activity of *Polygonatum sibiricum* Polysaccharides against D-Galactose-Induced Memory Impairment in Mice"

_ijms, 2022, doi:10.3390/ijms231911220_

Round 1

Reviewer 1 Report

Comments:

   In this manuscript, the authors described “P Use of steaming process to improve biochemical activity of Polygonatum sibiricum polysaccharides against D-galactose-induced memory impairment in mice”. This paper show that t steaming process improves the effects of Polygonatum sibiricum polysaccharides against D-gal-induced memory impairment in mice, likely through increasing the antioxidant activity of PSP.  However, there are a few points that need to be clarified.

Comment

1.     The preparation of PSP0, PSP1, PSP5, PSP9 and PSP15 must be clearly stated. The polysaccharide extraction method and recovery rate must also be clearly stated.

2.     What is the composition ratio of different monosaccharides in PSP0, PSP1, PSP5, PSP9 and PSP15 polysaccharides? Author should quantify it.

Reviewer 2 Report

The municipal script "Use of steaming process to improve biochemical activity of Polygonatum sibiricum polysaccharides against D-galactose-in-duced memory impairment in mice" is an attempt to improve the quality of life in old age

My comments:

Introduction; shorten the introduction. Try to describe only the substantive and describe the relationships between the analyzed parameters.

Material; why 15 animals per group (including the control group) were used for the experiment.

What was the insolence of the animals?

Whether the hippocampus for molecular testing was derived from one animal, eg ELISA method, Western blot.

Why the aging process was studied in 3-month-old mice. The young adult mouse is between 8-12 weeks old. The aging process is studied in approximately one-year-old animals.

What were the control animals?

Discussion; the description is based on changes in the aging process and studies have been conducted on young animals. Correct.

Correct linguistic mistakes.

Round 2

Reviewer 1 Report

ACCEPTED